# Production of the Green Leaf Volatile (*Z*)-3-Hexenal by a *Zea mays* Hydroperoxide Lyase

**DOI:** 10.3390/plants11172201

**Published:** 2022-08-25

**Authors:** Jessica P. Yactayo-Chang, Charles T. Hunter, Hans T. Alborn, Shawn A. Christensen, Anna K. Block

**Affiliations:** Chemistry Research Unit, Center for Medical, Agricultural and Veterinary Entomology, USDA-Agricultural Research Service, Gainesville, FL 32608, USA

**Keywords:** Arabidopsis, armyworm, defense, insect, maize, wounding, HPL, GLV

## Abstract

Plant-produced volatile compounds play important roles in plant signaling and in the communication of plants with other organisms. Many plants emit green leaf volatiles (GLVs) in response to damage or attack, which serve to warn neighboring plants or attract predatory or parasitic insects to help defend against insect pests. GLVs include aldehydes, esters, and alcohols of 6-carbon compounds that are released rapidly following wounding. One GLV produced by maize (*Zea mays*) is the volatile (*Z*)-3-hexenal; this volatile is produced from the cleavage of (9Z,11E,15Z)-octadecatrienoic acid by hydroperoxide lyases (HPLs) of the cytochrome P450 CYP74B family. The specific HPL in maize involved in (*Z*)-3-hexenal production had not been determined. In this study, we used phylogenetics with known HPLs from other species to identify a candidate HPL from maize (*ZmHPL*). To test the ability of the putative HPL to produce (*Z*)-3-hexenal, we constitutively expressed the gene in *Arabidopsis thaliana* ecotype Columbia-0 that contains a natural loss-of-function mutant in *AtHPL* and examined the transgenic plants for restored (*Z*)-3-hexenal production. Volatile analysis of leaves from these transgenic plants showed that they did produce (*Z*)-3-hexenal, confirming that ZmHPL can produce (*Z*)-3-hexenal in vivo. Furthermore, we used gene expression analysis to show that expression of *ZmHPL* is induced in maize in response to both wounding and the insect pests *Spodoptera frugiperda* and *Spodoptera exigua*. Our study demonstrates that ZmHPL can produce GLVs and highlights its likely role in (*Z*)-3-hexenal production in response to mechanical damage and herbivory in maize.

## 1. Introduction

Plants emit green leaf volatiles (GLVs) in response to damage and facilitate plant-to-plant communication by alerting neighboring plants to possible insect herbivory [1]. Plants exposed to GLVs (primed) upregulate jasmonic acid and downstream chemical defenses against herbivorous insects, leading to increased resiliency against insect attack [2]. In addition, the release of GLVs by wounded leaves functions in indirect defense by attracting insect predators and parasitoids of herbivores [3,4]; they can be used as host location signals by insect pests [5,6]; they can also protect seedlings from cold stress [7], suppress pathogen growth [8], and prime plants for accelerated growth [9].

An important part of the function of GLVs is their latent biosynthetic pathway; this feature enables the rapid production of GLVs within minutes of tissue damage and prevents high levels of the compounds from accumulating within the plant tissues [10]. A large body of work exists on the biosynthesis and varied functions of GLVs in plants, for reviews on this topic see [11,12,13,14,15].

GLV blends include six carbon aldehydes, alcohols, and esters, which, as their name suggests, are responsible for the “grassy” smell of cut grass. GLVs are oxylipins formed by lipid peroxidation in the chloroplasts; their synthesis begins with lipoxygenase activity, which catalyzes the conversion of the fatty acid linolenic acid into the hydroperoxide, (9Z,11E,15Z)-octadecatrienoic acid (13-HPOT) (Figure 1) [16]. 13-HPOT can then be cleaved by hydroperoxide lyase (HPL) to release the first GLV, the six-carbon aldehyde (*Z*)-3-hexenal; this can then be enzymatically converted to other GLVs, including (3Z)-hexenol and (3Z)-hexenyl acetate. 13-HPOT also serves as the substrate for allene oxide synthase (AOS), upstream of jasmonic acid (JA) biosynthesis [17]. Competition for 13-HPOT between HPL and AOS is lessened by the pairing of dedicated 13-LOXs and HPL or AOS in multimeric protein complexes that directly feed each pathway [18].

In maize, the 13-lipoxygenase, LOX10 (GRMZM2G015419) is the dedicated 13-LOX responsible for 13-HPOT production in GLV synthesis, and loss of function mutants produce little to no GLVs [19]; however, the corresponding HPL that works with LOX10 for the conversion of 13-HPOT into (*Z*)-3-hexenal has not been identified. Functionally validated HPLs from other plant species are members of the CYP74B subgroup of the CYP74 family of cytochrome p450s. The CYP74 family is involved in the metabolism of fatty acid hydroperoxides to produce diverse bioactive oxylipins in plants. Other subgroups of this family have AOS (CYP74A and CYP74C) and divinyl ether synthase (CYP74D) activity [20,21]. Active HPLs have been characterized in several plant species including Arabidopsis (*Arabidopsis thaliana*), tomato (*Solanum lycopersicum*) and rice (*Oryza sativa*) [22,23,24,25].

In this study we use phylogenetic analysis with known HPLs to identify a candidate HPL from maize. We then functionally validate the ability of this enzyme to produce (*Z*)-3-hexenal in vivo by constitutively expressing it in an HPL-deficient Arabidopsis line. Furthermore, since GLVs are induced in response to damage and herbivory we investigate the expression of the candidate *HPL* gene in maize in response to wounding, caterpillar oral secretions, and herbivory by the maize pest insects *Spodoptera frugiperda* (fall armyworm) and *Spodoptera exigua* (beet armyworm).

## 2. Results

### 2.1. Phylogenetic Analysis Reveals That ZmHPL1 Clusters with Known HPLs from Other Species

Phylogenetic analysis revealed well-supported subgroups within the CYP74 families in Arabidopsis, tomato, rice, and maize (Figure 2). The predicted maize HPL (*ZmHPL1*, GRMZM6G986387) falls within the CYP74B subfamily and groups closely with validated HPL enzymes; *AtHPL1* from Arabidopsis [22,23], *SolycHPL* from tomato [24,26], and *OsHPL3* from rice [25,27]. Therefore, we hypothesized that the maize gene *ZmHPL1* is the best candidate for encoding an active HPL in maize. Another maize gene, which we named *ZmHPL-like1* (GRMZM2G168404), also falls within the CYP74B subfamily but groups with the rice genes *OsHPL1* and *OsHPL2*, which do have HPL activity, though with lower substrate specificity and broad expression patterns [25]. Five maize gene products group within Cyp74A with known AOS enzymes from Arabidopsis, tomato, and rice. No maize, rice, or Arabidopsis gene products fall within Cyp74C and Cyp74D subfamilies.

### 2.2. ZmHPL1 Restores (Z)-3-Hexenal Production in Arabidopsis AtHPL1 Loss-of-Function Plants

To determine if ZmHPL1 is competent to produce (*Z*)-3-hexenal, the gene was cloned from maize and constitutively expressed in the Arabidopsis ecotype Col-0; this ecotype contains a 10-nucleotide deletion in the first exon of *AtHPL1*, resulting in a truncated non-functional HPL [23]. Three transgenic lines (L1, L2 and L3) were isolated and expression analysis revealed that all three lines expressed *ZmHPL1* (Figure 3a). To determine whether this expression led to the production of (*Z*)-3-hexenal, GLVs were measured from ground leaf tissue from the three transgenic lines and the Col-0 control (Figure 3b). The Arabidopsis ecotypes Ler-0 and Ws-0 have functional *AtHPL1* [23] and were used as positive controls for (*Z*)-3-hexenal production in Arabidopsis. Two of the ZmHPL1 lines (L1 and L3) had significantly higher (*Z*)-3-hexenal production than the trace levels observed in Col-0 (Figure 3b). The third line (L2) also had elevated levels, though the significant difference from Col-0 was less pronounced (*p* = 0.07 by *t*-test). Levels of (*Z*)-3-hexenal in the ZmHPL lines were comparable to those seen in Ler-0 and Ws-0 plants; these data show that ZmHPL1 can complement the loss-of-function of AtHPL1 in Col-0 to restore HPL activity. 

### 2.3. Wounding and Herbivory Induce ZmHPL1 Expression in Maize

To determine the impact of herbivory on *ZmHPL1* expression in maize, the expression of *ZmHPL1* was measured in untreated maize leaves and leaves of plants infested for 24 h with neonates of *Spodoptera frugiperda* (fall armyworm) or *Spodoptera exigua* (beet armyworm). The expression of *ZmHPL1* was significantly induced after 24 h infestation with either *S. frugiperda* or *S. exigua*, increasing around 40-fold and 15-fold, respectively, compared to untreated controls (Figure 4a). To determine the impact of wounding and larval oral secretions on *ZmHPL* expression, a 2 cm^2^ area on either side of the leaf midvein was gently scored with a razor blade (wounding) or scored with a razor blade and treated with *S. frugiperda* oral secretions. Tissue samples were collected on untreated leaves (0 min) and at 5, 20, and 60 min following treatment and analyzed for *ZmHPL* expression (Figure 4b). Wounding alone induced a slight (1.5-fold) but significant increase in *ZmHPL* expression at 60 min after treatment. Wounding plus oral secretion treatment also induced a slight (2-fold) but significant increase in *ZmHPL* expression and this response was more rapid than that of wounding alone as a significant difference from untreated controls was observed at 20 min after treatment; these data indicate that herbivory does induce the expression of *ZmHPL* in maize.

## 3. Discussion

We used phylogenetic analyses to identify a predicted HPL from maize, *ZmHPL,* that grouped closely with known HPLs from rice, tomato and Arabidopsis (Figure 2). Heterologous expression of the putative ZmHPL was able to complement the HPL deficiency of the Arabidopsis Col-0 ecotype, enabling it to produce (*Z*)-3-hexenal at levels equivalent to those of the Arabidopsis ecotypes Ler-0 and Ws-0 that have functional AtHPL (Figure 2). *ZmHPL* therefore encodes a functional HPL in vivo. 

Expression analysis revealed that *ZmHPL* was induced in maize one hour after a one-time mechanical wounding event (Figure 3). The addition of *S. frugiperda* oral secretions to this one-time wounding treatment led to a more rapid induction of *ZmHPL* expression; these data suggest that both mechanical damage and recognition of a component of boiled *S. frugiperda* oral secretions can lead to the induction of *ZmHPL*. The expression of *ZmHPL* was also induced in maize plants infested with either *S. frugiperda* or *S. exugia* larvae for 24 h. As GLVs in maize are produced in response to damage and have important roles in defense against herbivores such as *S. frugiperda* by attracting parasitoids [19], our expression data support the hypothesis that ZmHPL functions to sustain herbivory-induced GLV production in maize.

Wounding and herbivory have been shown to impact the expression of HPL in other plant species. For instance, wounding induces the expression of *AtHPL* in Arabidopsis [28] and *NaHPL* in *Nicotine attenuata* [29]. On the other hand, the impact of insect oral secretions on *HPL* expression appears to be species-specific, as regurgitate from *Manduca sexta* suppressed wound-induced *NaHPL* expression [29] while no significant impact on *AtHPL* expression was observed in Arabidopsis during treatment with *Pierse rapae* regurgitate, even though GLV emissions were suppressed [30]. Importantly, control of GLV biosynthesis in the short term is not likely to be predominantly by transcriptional regulation of *HPL* [20] but rather by substrate flux that is limited by the release of 13-HPOT [31]; this hypothesis is supported by the observation that Arabidopsis plants (Ecotype No-0) overexpressing *HPL* from pepper do not display constitutive increases in GLV emission [32].

A meta-analysis of existing studies on GLVs showed that the amount of GLVs produced by plants is dependent on the stresses it encounters, with fungal infection leading to the highest induction amongst biotic stress treatments tested, followed by wounding and then herbivory [13]. Maize has been shown to produce GLVs in response to wounding or herbivory by insects such as *S. exguia* [19]; however, insects have active defenses against GLVs and produce GLV suppressing effectors. For example, the oral secretions of *S. frugiperda* include an isomerase that converts (*Z*)-3-hexenal to (E)-2-hexenal; a fatty acid dehydratase which converts 13-HPOT to (9Z,11E)-13-oxo-octadecadienoic acid (13-ODE), preventing the biosynthesis of (*Z*)-3-hexenal and subsequent GLVs; and a heat-stable hexenal trapping (HALT) molecule that specifically binds to (*Z*)-3-hexenal preventing GLV release [33]. In comparison, the oral secretions of *S. exguia* contain only the HALT effector [33]. Treatment with the oral secretions of either *S. frugiperda* or *S. exguia* therefore suppresses (*Z*)-3-hexenal production in maize compared to wound alone controls [34]; this can explain the disconnect between the increase in *ZmHPL* expression due to *S. frugiperda* and *S. exguia* oral secretion observed in this study and the known reduction in GLV production caused by oral secretion treatment is likely due to the impact of the oral secretions effectors of *S. frugiperda* and *S. exguia.*

To conclude, our study shows that *ZmHPL* encodes a functional HPL of maize and is likely involved in herbivore-induced GLV production; however, whether this gene encodes the only functional HPL enzyme in maize, and what its relative contribution to GLV production is under various stress conditions, remain to be answered using loss-of-function studies.

## 4. Materials and Methods

### 4.1. Phylogenetic Analyses

Predicted protein sequences for the Cyp74 gene family were identified from BLASTP queries of gene databases for maize (Ensembl-18), Arabidopsis (TAIR10), rice (v7_JGI), and tomato (iTAG2.4) using the Arabidopsis HPL (AF087932) from *Landsberg erecta (Ler-0)*. All queries were conducted on Phytozome 12 (JGI). Cutoff for inclusion in the phylogenetic analysis was set at e-values greater than 1 × 10^−50^. The putative maize HPL (GRMZM6G986387) was incorrectly annotated in previous maize gene sets (including Ensembl-18), being oppositely oriented and thus not identified in BLASTP queries using HPL or AOS sequences. The PH207v1.1 assembly was also queried using AtHPL and ZmHPL1 was identified and included in phylogenetic analyses along with the HPL-related genes from Ensembl-18. Altogether, two genes from Arabidopsis, five from rice, 7 from tomato, and 7 from maize were identified as belonging to the CYP74 family. A MUSCLE alignment of predicted protein sequences from the primary transcript of each gene was conducted within the Geneious software interface (Biomatters Limited, Auckland, New Zealand). The Phylogeny was determined using the Neighbor-joining method based on the Jukes-Cantor model with no outgroup and 1000 random-seeded bootstrap repetitions. Accession numbers and gene abbreviations for the predicted proteins can be found in Appendix A.

### 4.2. Cloning and Transformation of ZmHPL into Arabidopsis

The full-length *ZmHPL1* was amplified from maize genomic DNA from the inbred line B73 using PCR with primers 5′-caccATGCTGCCGTCCTTCGTGTCGCCGAC-3′ and 5′-CTGCTGCGCTCCGGCGGCTGCTGC-3′ and transferred into the gateway entry vector pENTR-D/Topo (Invitrogen, Waltham, MA, USA). As *ZmHPL1* contains no introns, amplification from genomic DNA resulted in cloning the predicted full-length cDNA. The primers were designed to lack the stop codon and the subsequent transfer of the cDNA to the gateway destination vector pK7FWG2 [35] led to the in-frame fusion of *ZmHPL1* to a C-terminal GFP tag under the control of a constitutive 35S promotor. The pK7FWG2:*ZmHPL1* plasmid was then placed in *Agrobacterium tumefaciens* and used to transform Arabidopsis Columbia-0 plants via the floral dip method [36]. Positive transformants were selected on 1/2 x Murashige and Skoog agar plates containing kanamycin.

### 4.3. Plant Growth Conditions and Treatments

Surface sterilized Arabidopsis seeds were grown on 1/2 x Murashige and Skoog agar plates with kanamycin for the *ZmHPL1* lines and without kanamycin for the Columbia (Col-0), Landsberg *erecta* (Ler-0) and Wassilewskija (Ws-0) lines. At the emergence of the first true leaves, and after observation of kanamycin sensitivity in non-transgene-containing segregants, the seedlings were transferred into the soil. Arabidopsis plants were grown in a growth chamber with 12 h day/night cycles at 25 °C. Gene expression and (*Z*)-3-hexenal analysis were performed on fully expanded leaves of 4-week-old plants. Four biological replicates were conducted per line.

### 4.4. ZmHPL Gene Expression Analyses

Total RNA was extracted from Arabidopsis or maize leaf tissue using the Plant RNeasy™ mini kit with on-column DNase treatment (Qiagen, Hilden, Germany) following the manufacturers protocols. The RNA was reverse transcribed into cDNA using RETROscript™ Reverse Transcription Kit (Ambion, Austin, TX, USA) with an oligo dT primer following the manufacturers protocols. Relative expression levels were determined by quantitative Real-Time PCR (qRT-PCR) using SSoAdvanced Universal SYBR Green Supermix^®^, (Biorad, Hercules, CA, USA) using the geometric mean of reference genes folylpolyglutamate synthase (*fpgs*, GRMZM2G393334) and a ubiquitin carrier protein (*ubcp*, GRMZM2G102471) for maize [37] and *actin2 (At3g18780)* for Arabidopsis [38] according to the 2^(−ΔΔcq)^ method. The gene-specific primers 5′-CAGCATGTTGTTGATGGCGT-3′ and 5′-AGCTGCTCATCCACTCGTTC-3′ were used to determine *ZmHPL* expression.

### 4.5. (Z)-3-Hexenal Quantification

(*Z*)-3-hexenal measurements were conducted as in [39]. Briefly, 20 mg of leaf tissue from fully expanded leaves was collected and ground using a beadbeater in the presence of 25 μL 1x PBS buffer (pH 7) and frozen in liquid nitrogen. Headspace volatiles were collected from the samples immediately following grinding by pulling a vacuum through SuperQ volatile collection filters at 100 mL·min^−1^ for 5 min as samples thawed. Filters were eluted using 150 µL dichloromethane containing 400 ng 2-heptanal (used as an internal standard) and (*Z*)-3-hexenal was quantified on an Agilent (Santa Clara, CA, USA) 7890GC/4000B TSQ MS in EI mode (70 EV and ion source temperature set to 220 °C). One μL of the eluted sample was injected using on-column injection. The injector was set to follow oven mode (programmed to stay 3 °C above oven temperature) and equipped with a 0.5-m deactivated retention gap attached to the analytical column using a fused silica connector (Supelco, Bellefonte, PA, USA). The 30 m, 255 mmID, 025 μm film thickness DB-5 column (Agilent) was kept at 30 °C for 2 min followed by 5 °C·min^−1^ to 80 °C and then a rapid increase to 260 °C to clean the column between injections. Typically, only two peaks were visible during the first 10 min of the analyses, (*Z*)-3-hexenal and 2-heptanal. (*Z*)-3-hexenal was confirmed by standard injections and by comparison to the NIST14 GC/MS library. Four biological replicates were conducted per line or treatment.

### 4.6. Statistical Analyses

For determination of statistical significance, *t*-tests were used to compare treatments against appropriate controls. For quantifications of Z-3-hexenal, each treatment condition was compared to Col-0. For gene expression analysis, the relative quantities were calculated using the 2^−ΔΔCt^ method, and statistical tests were run on the transformed data. Relative quantity was compared against untreated controls or time 0. A value of *p* ≤ 0.05 was considered statistically significant. Each experiment was conducted two times with similar results and the data from the second set of experiments is presented.

## Figures and Tables

**Figure 1 plants-11-02201-f001:**
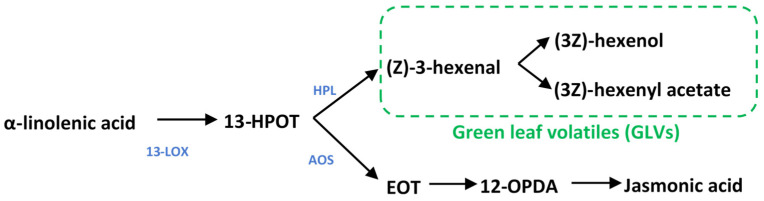
**Metabolism of green leaf volatiles and jasmonates.** 13-lipoxygenase converts a-linolenic acid into 13-HPOT, which can then be diverted into GLV production by HPL or jasmonate synthesis by AOS and downstream enzymes. 13-LOX13: 13-lipoxygenase; HPL: hydroperoxide lyase; AOS: allene oxide synthase; 13-HPOT: (9Z11E15Z13S)-hydroperoxyl-9,11,15-octadecantrienoic acid; EOT: 12,13(S)-epoxy-9(Z)-octadecatrienoic acid; 12-OPDA: cis-(+)-12-oxophytodienoic acid.

**Figure 2 plants-11-02201-f002:**
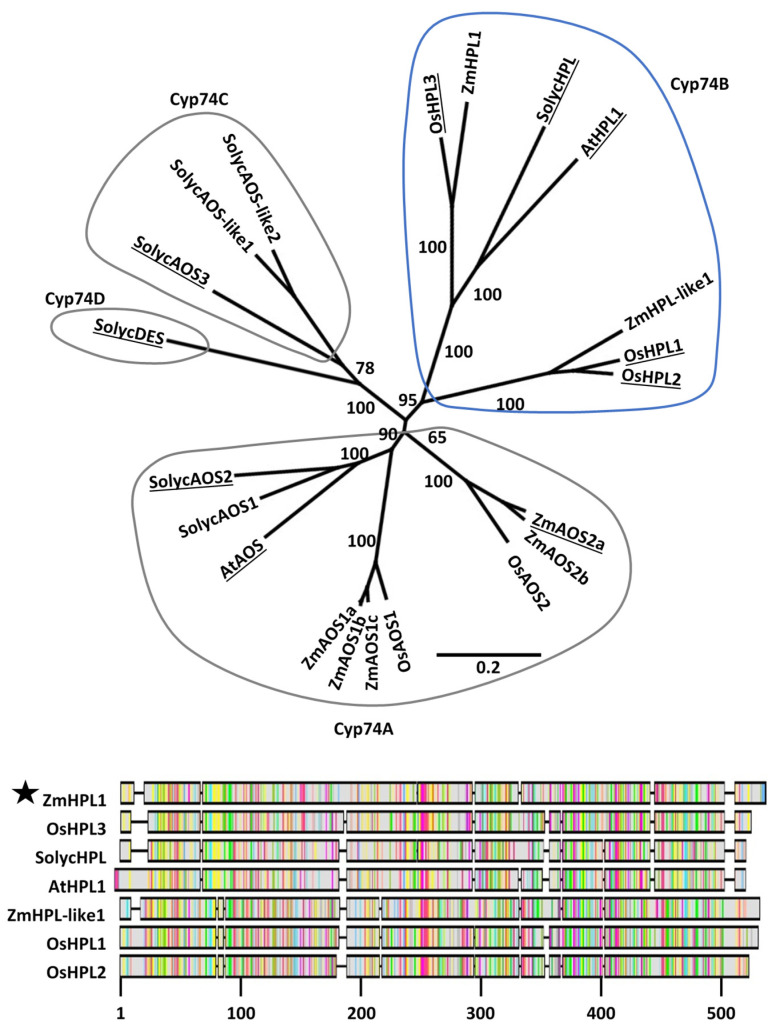
**Phylogenetic analysis of the Cyp74 gene families of Arabidopsis, tomato, rice, and maize.** The Neighbor-joining tree was constructed using MUSCLE alignments of predicted protein sequences based on primary transcripts of each gene. Enzymes for which biochemical activity has been experimentally determined are underlined. Cyp74 subfamilies are circled. ZmHPL1 falls within the Cyp74B subfamily and is closely associated with biochemically validated HPLs. Relevant bootstrap values (1000 repetitions) are shown. The scale bar indicates substitutions per site. The alignment includes predicted protein sequences of the seven Cyp74B members of maize, rice, tomato, and Arabidopsis. The most highly conserved regions of amongst Cyp74B members are between positions 40 and 110 and from positions 310 to 490.

**Figure 3 plants-11-02201-f003:**
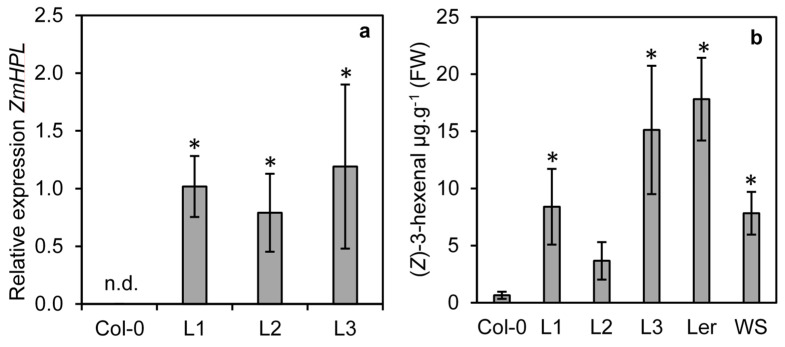
**Expression of *ZmHPL* in HPL deficient Arabidopsis ecotype Col-0 restores (*Z*)-3-hexenal production.** The Col-0 ecotype of Arabidopsis (a natural loss-of-function mutant in *AtHPL*) was agrotransformed to constitutively express *ZmHPL*. Three lines from independent transformation events L1, L2 and L3 were assessed for their expression of *ZmHPL* using qRT-PCR, n.d. = not detected (**a**). Production of (*Z*)-3-hexenal was measured from Col-0, the three transgenic lines and two Arabidopsis ecotypes that have wild-type AtHPL activity (Ler and WS) (**b**). Bars show the average of n = 4 samples per line, ±S.E.M. Asterisks indicate significantly different values from Col-0 at *p* ≤ 0.05 as determined by a *t*-test.

**Figure 4 plants-11-02201-f004:**
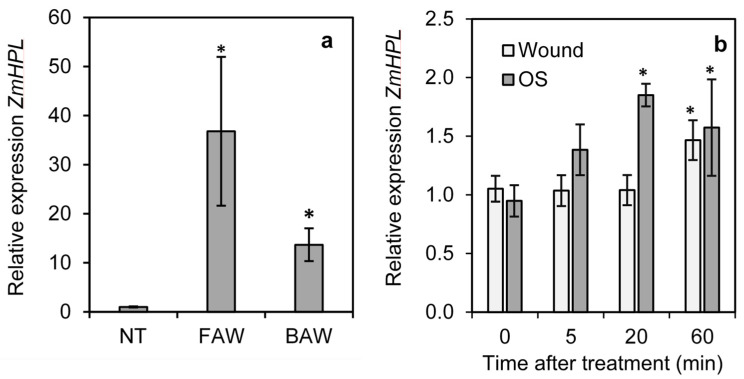
**Impact of wounding and *S. frugiperda* or *S. exigua* feeding on the expression of *ZmHPL*.** The relative expression levels of *ZmHPL* were determined in wild-type maize plants infested with *S. frugiperda* neonates (FAW) or *S. exigua* neonates (BAW) for 24 h or not treated (NT) (**a**); or in maize plants wounded with a razorblade or wounded and treated with *S. frugiperda* oral secretions (OS) (**b**). Bars show the average of n = 4 samples per line, ±S.E.M. Asterisks indicate relative expression significantly different from NT or 0 time at *p* ≤ 0.05 by *t*-test.

## Data Availability

All data is available upon request from the corresponding author.

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
