# Peer review of "Production of the Green Leaf Volatile (Z)-3-Hexenal by a Zea mays Hydroperoxide Lyase"

_plants, 2022, doi:10.3390/plants11172201_

Round 1
Reviewer 1 Report
GLVs include aldehydes, esters, and alcohols of 6-carbon compounds that are released rapidly following wounding, which play a crucial role in the indirect defense of plants. Little is known about the maize HPLs involved in (Z)-3-hexenal production. This MS focused on biochemical confirmation of the function of a ZmHPL in catalyzing the production of (Z)-3-hexenal. The authors expressed the target gene in Arabidopsis Col-0, which is a natural mutant of AtHPL, and found leaves of these transgenic plants produced (Z)-3-hexenal, confirming that ZmHPL can produce (Z)-3-hexenal in vivo. Furthermore, expression of ZmHPL is induced in response to wounding and infestation by Spodoptera frugiperdaand Spodoptera exigua in maize. These findings reveal that ZmHPL can produce GLVs and may play a role in maize defense response.
1. In Fig 2, a colored protein sequence alignment would be better.
2. In Fig 4a, the size of asterisk on top of the bar of BAW should be changed.
3. It would be better if the authors could provide the emission of (Z)-3-hexenal in maize plants infested with Spodoptera frugiperda and Spodoptera exigua and/or treated with wounding and OS.
4. Line 80, for OsHPL3, an in vivo study in rice (Tong et al. Plant Journal, 2012, 71:763-775) should be cited as well.
5. Lines 220 and 223, 1/2 × Murashige and Skoog (times sign). Do not use the capital letter X.
Author Response
Response to Reviewer 1
Reviewer 1
Comments and Suggestions for Authors
GLVs include aldehydes, esters, and alcohols of 6-carbon compounds that are released rapidly following wounding, which play a crucial role in the indirect defense of plants. Little is known about the maize HPLs involved in (Z)-3-hexenal production. This MS focused on biochemical confirmation of the function of a ZmHPL in catalyzing the production of (Z)-3-hexenal. The authors expressed the target gene in Arabidopsis Col-0, which is a natural mutant of AtHPL, and found leaves of these transgenic plants produced (Z)-3-hexenal, confirming that ZmHPL can produce (Z)-3-hexenal in vivo. Furthermore, expression of ZmHPL is induced in response to wounding and infestation by Spodoptera frugiperda and Spodoptera exigua in maize. These findings reveal that ZmHPL can produce GLVs and may play a role in maize defense response.
- In Fig 2, a colored protein sequence alignment would be better.
We have replaced the black and white alignment with a colored version. Thank you for the suggestion.
- In Fig 4a, the size of asterisk on top of the bar of BAW should be changed.
corrected
- It would be better if the authors could provide the emission of (Z)-3-hexenal in maize plants infested with Spodoptera frugiperda and Spodoptera exigua and/or treated with wounding and OS.
Maize has been shown to produce GLVs in response to wounding or herbivory by insects such as S. exguia with oral secretions from caterpillars suppressing GLV production (Jones et al 2019,2022 etc). The phenomenon that maize produces higher levels of GLVs upon wounding than following herbivory is well established. Furthermore, the OS treatment increases HPL expression, yet because of OS effectors decreases GLV production compared to wounding. To clarify this issue we have expanded the discussion section to include what is known about herbivory on GLV production in maize and the impact of OS effectors on these processes.
- Line 80, for OsHPL3, an in vivo study in rice (Tong et al. Plant Journal, 2012, 71:763-775) should be cited as well.
Thank you, the citation was added.
- Lines 220 and 223, 1/2 × Murashige and Skoog (times sign). Do not use the capital letter X.
corrected

Reviewer 2 Report
This manuscript is well-written and provides compelling evidence for the role of ZmHPL in (Z)-3-hexenal production in maize. The authors conducted complementary phylogenetic, return of function, expression, and volatile production experiments to determine that ZmHPL plays a key role in GLV production in maize.
Line 186 missing “is”
Author Response
Response to Reviewer 2
Comments and Suggestions for Authors
This manuscript is well-written and provides compelling evidence for the role of ZmHPL in (Z)-3-hexenal production in maize. The authors conducted complementary phylogenetic, return of function, expression, and volatile production experiments to determine that ZmHPL plays a key role in GLV production in maize.
Thank you for your review.
Line 186 missing “is”
corrected

Reviewer 3 Report
Review ID 1866052
Production of the Green Leaf Volatile (Z)-3-hexenal by a Zea 2 mays Hydroperoxide Lyase
This is very well organized and written paper. It was my pleasure to review this manuscript.
Critical review:
1. 1. Introduction must be corrected. Z-3-hexenal is probably one of the most important compound/GLV associated with plant’s respond to stress. However, this is only one of the green leaf volatiles. What about the other GLVs? You can find many papers dealing with volatiles. So, there is not enough information about volatile organic compounds which play a crucial role as plant’s defiance system. Some papers below.
2. Sections 4.5 and 4.6 are poorly described.
3. Discussion in present form can’t be accepted. Again, try to find more papers dealing with GLVs.
4. What about conclusions? 2-3 sentences are of importance for the readers. Please do it.
Some other paper for consideration:
Tribolium confusum responses to blends of cereal kernels and plant volatiles
Journal of Applied Entomology 140, 558–563 (2016)
DOI: 10.1111/JEN.12284
Sitophilus granarius responses to blends of five groups of cereal kernels and one group of plant volatiles
Journal of Stored Products Research 62: 36-39 (2015)
DOI: 10.1016/J.JSPR.2015.03.007
Author Response
Response to Reviewer 3
Comments and Suggestions for Authors
Review ID 1866052
Production of the Green Leaf Volatile (Z)-3-hexenal by a Zea 2 mays Hydroperoxide Lyase
This is very well organized and written paper. It was my pleasure to review this manuscript.
Critical review:
- Introduction must be corrected. Z-3-hexenal is probably one of the most important compound/GLV associated with plant’s respond to stress. However, this is only one of the green leaf volatiles. What about the other GLVs? You can find many papers dealing with volatiles. So, there is not enough information about volatile organic compounds which play a crucial role as plant’s defiance system. Some papers below.
We thank the reviewers for their suggestion and have added the recommended papers and added detail to the introduction and added references to some current reviews on the broad topic of GLV function. We feel that details of the role of non GLV volatiles, although important, are beyond the scope of this short communication which is centered on the identification of an HPL in maize rather than the biological function of GLVs in general.
- Sections 4.5 and 4.6 are poorly described.
Detail was added to clarify sections 4.5 and 4.6.
- Discussion in present form can’t be accepted. Again, try to find more papers dealing with GLVs.
As requested, we have added to the discussion on GLV’s specifically information relating to GLV production in maize and the impact of S. frugiperda and S. exguia effectors on GLV suppression.
- What about conclusions? 2-3 sentences are of importance for the readers. Please do it.
Our conclusions are at the end of the discussion (last paragraph of section 3 of this short communication) “To conclude, our study shows that ZmHPL encodes a functional HPL of maize and is likely involved in herbivore induced GLV production.”
Some other paper for consideration:
Tribolium confusum responses to blends of cereal kernels and plant volatiles
Journal of Applied Entomology 140, 558–563 (2016)
DOI: 10.1111/JEN.12284
Added as requested
Sitophilus granarius responses to blends of five groups of cereal kernels and one group of plant volatiles
Journal of Stored Products Research 62: 36-39 (2015)
DOI: 10.1016/J.JSPR.2015.03.007
Added as requested

Round 2
Reviewer 3 Report
Thank you.